# Proteogenomics Reveals Orthologous Alternatively Spliced Proteoforms in the Same Human and Mouse Brain Regions with Differential Abundance in an Alzheimer’s Disease Mouse Model

**DOI:** 10.3390/cells10071583

**Published:** 2021-06-23

**Authors:** Esdras Matheus Gomes da Silva, Letícia Graziela Costa Santos, Flávia Santiago de Oliveira, Flávia Cristina de Paula Freitas, Vinícius da Silva Coutinho Parreira, Hellen Geremias dos Santos, Raphael Tavares, Paulo Costa Carvalho, Ana Gisele da Costa Neves-Ferreira, Andrea Siqueira Haibara, Patrícia Savio de Araujo-Souza, Adriana Abalen Martins Dias, Fabio Passetti

**Affiliations:** 1Instituto Carlos Chagas, FIOCRUZ, Rua Professor Algacyr Munhoz Mader 3775, Cidade Industrial De Curitiba, Curitiba, PR 81310-020, Brazil; esdrassilva@aluno.fiocruz.br (E.M.G.d.S.); lcosta@aluno.fiocruz.br (L.G.C.S.); flavia.paula@fiocruz.br (F.C.d.P.F.); vparreira@aluno.fiocruz.br (V.d.S.C.P.); hellen.santos@fiocruz.br (H.G.d.S.); paulo@pcarvalho.com (P.C.C.); 2Laboratory of Toxinology, Oswaldo Cruz Institute (FIOCRUZ), Av. Brazil 4365, Manguinhos, Rio de Janeiro, RJ 21040-900, Brazil; anag@ioc.fiocruz.br; 3Laboratório de Inflamação e Câncer, Departamento de Genética, Ecologia e Evolução, Instituto de Ciências Biológicas, Universidade Federal de Minas Gerais (UFMG), Avenida Presidente Antônio Carlos 6627, Pampulha, Belo Horizonte, MG 31270-901, Brazil; flaviaiso@ufmg.br (F.S.d.O.); abalen@ufmg.br (A.A.M.D.); 4Departamento de Bioquímica e Imunologia, Instituto de Ciências Biológicas, Universidade Federal de Minas Gerais (UFMG), Avenida Presidente Antônio Carlos 6627, Pampulha, Belo Horizonte, MG 31270-901, Brazil; tawares07@gmail.com; 5Departamento de Fisiologia e Biofísica, Instituto de Ciências Biológicas, Universidade Federal de Minas Gerais (UFMG), Avenida Presidente Antônio Carlos 6627, Pampulha, Belo Horizonte, MG 31270-901, Brazil; haibara@icb.ufmg.br; 6Laboratory of Immunogenetics and Histocompatibility, Department of Genetics, Universidade Federal do Paraná, Av. Cel. Francisco H. dos Santos 100, Jardim das Américas, Curitiba, PR 81530-980, Brazil; psas@ufpr.br

**Keywords:** Alzheimer’s disease, mRNA splicing, neurodegeneration, proteogenomics

## Abstract

Alternative splicing (AS) may increase the number of proteoforms produced by a gene. Alzheimer’s disease (AD) is a neurodegenerative disease with well-characterized AS proteoforms. In this study, we used a proteogenomics strategy to build a customized protein sequence database and identify orthologous AS proteoforms between humans and mice on publicly available shotgun proteomics (MS/MS) data of the corpus callosum (CC) and olfactory bulb (OB). Identical proteotypic peptides of six orthologous AS proteoforms were found in both species: PKM1 (gene *PKM/Pkm*), STXBP1a (gene *STXBP1/Stxbp1*), Isoform 3 (gene *HNRNPK/Hnrnpk*), LCRMP-1 (gene *CRMP1/Crmp1*), SP3 (gene *CADM1/Cadm1*), and PKCβII (gene *PRKCB/Prkcb*). These AS variants were also detected at the transcript level by publicly available RNA-Seq data and experimentally validated by RT-qPCR. Additionally, PKM1 and STXBP1a were detected at higher abundances in a publicly available MS/MS dataset of the AD mouse model APP/PS1 than its wild type. These data corroborate other reports, which suggest that PKM1 and STXBP1a AS proteoforms might play a role in amyloid-like aggregate formation. To the best of our knowledge, this report is the first to describe PKM1 and STXBP1a overexpression in the OB of an AD mouse model. We hope that our strategy may be of use in future human neurodegenerative studies using mouse models.

## 1. Introduction

Proteogenomics is an emerging field that integrates proteomic, transcriptomic, and genomic data to assess the multiple regulatory levels of gene expression within the cell [1]. The identification of peptide sequences through tandem mass spectrometry (MS/MS) is often achieved by matching experimental and theoretical mass spectra using a standard protein sequence database [2]. However, a set of high-quality mass spectra can remain unidentified [3]. Integrating data from multiple levels, such as the genome and transcriptome, permits the construction of customized protein sequence databases that can improve peptide identification in unassigned mass spectra [4]. Most of these unassigned mass spectra correspond to proteoforms derived from molecular events, such as post-translation modifications (PTMs) and alternative splicing (AS), which are usually absent in standard protein sequence databases [5].

AS consists of the alternative usage of pre-mRNA exons and introns, producing more than one mRNA from a single gene [6,7]. Approximately 95% of human pre-mRNAs are alternatively spliced [8], with the resulting mRNA variants regulated in response to tissue- and developmental stage-specific factors [8]. When translated, these AS variants can generate distinct proteoforms, increasing proteome diversity [9]. The human brain contains a large diversity of AS variants, which can be detected at the transcript [10] and protein [11] levels. These variants play roles in neuronal processes such as neurogenesis [12], neuronal receptor activation [13], functional specification of neuronal cell types and circuits [14], neuronal migration [15], neuronal excitation [16], and synaptogenesis [17], as well as in neurodegenerative diseases such as Alzheimer’s disease (AD) [18].

It is also known that AS patterns tend to be more conserved in the mammalian nervous system [19]. Mouse models have been widely used to characterize different regions of the central nervous system, including the corpus callosum (CC) [20] and olfactory bulb (OB) [21]. CC shape [22,23] and size [24,25,26,27,28] are commonly altered in the brains of AD patients, and the OB is also known to be affected [29,30,31,32], indicating the importance of these brain regions for understanding the onset and progression of AD. Moreover, these results can only be extrapolated to human biology with a proper understanding of the molecular similarities and differences between human and mouse brain tissues [33].

The usefulness of existing MS/MS data in comparative biology is highlighted by the wealth of MS/MS datasets of different organisms available in public repositories [34]. In this study, we used a computational framework to identify peptides from publicly available MS/MS data, which led to the characterization of novel AS proteoforms conserved in the human and mouse brain. Some of these proteoforms were more abundant in OB MS/MS data of an AD murine model (*APP/PS1*). The novel AS proteoforms were confirmed in publicly available RNA-Seq data and experimentally validated by RT-qPCR in the CC and OB of mouse samples. We also provide a customized protein sequence database of humans and mice with novel AS proteoforms.

## 2. Materials and Methods

### 2.1. Protein Sequence Database Setup

First, we applied the SpliceProt method described by Tavares et al. (2014) [35] to identify AS transcripts from the Ensembl database (version 100) [36]. Additionally, we removed AS variants that may be potential targets for the nonsense-mediated decay (NMD) pathway by identifying variants with a premature termination codon using the software NMD Classifier [37]. These sequences were then translated in silico and combined with UniProt/Swiss-Prot protein sequences (version 12_2020) [38] to build the custom protein sequence database, following the method described by Tavares et al. (2017) [39]. Briefly, the sequences were divided into canonical and noncanonical proteoforms, and the noncanonical proteoforms were digested in silico by trypsin using a modified version of the Digest software (EMBOSS package version 6.3.1) [40]. The resulting non-redundant peptide sequences were compared to the canonical protein sequences such that only those exclusive to AS noncanonical proteoforms remained. Peptides that were not present in any canonical sequence were appended to a FASTA format file containing UniProt/SwissProt canonical protein sequences. According to the UniProt Consortium, canonical proteoforms are prevalent and similar to orthologous sequences, with the clearest description of domains, polymorphisms, and post-translational modification [41]. All the scripts and protein sequence databases used to build the human and mouse customized protein sequence databases can be accessed at https://github.com/Matheusdras/Make_Custom_DB, accessed on 26 March 2021.

### 2.2. Database Search

MS/MS datasets of healthy brain tissues were retrieved from the PRIDE repository of the ProteomeXchange Consortium (Table 1). An additional dataset with isobaric labeled peptides (iTRAQ-8) of an AD mouse model (*APP/PS1*) and its wild-type control was retrieved to investigate relative peptide abundance levels in OB (identifier PXD003312) [42]. All RAW data were analyzed using PatternLab for Proteomics v.5 software [43] and a specific custom sequence database for each species. The precursor mass tolerance used was 10, the maximum number of allowed modifications per peptide and missed cleavages was 2, and the enzyme specificity was set as semi-specific. For human data from healthy brain tissues, carbamidomethylation of cysteine was set as fixed, and oxidation of methionine and carbamidomethylation of DEHK were set as variable modifications. For the data from healthy mouse brain tissue, carbamidomethylation of cysteine was set as fixed, and oxidation of methionine and deamidation of NQ were set as variable modifications. We applied a 1% false discovery rate (FDR) at the protein level to converge to a confident list of identifications.

For the isobaric labeled *APP/PS1* and wild-type data, carbamidomethylation of C and iTRAQ8-plex labeling for K and the N-terminus were set as fixed, and phosphorylation on S/T/Y and deamidation of N were set as variable modifications. The iTRAQ data were quantified using the isobaric labeling feature of PatternLab for Proteomics with the following parameters: marker PPM equals 100, ion count threshold equals 0.025, and normalization based on identified spectra. The results were filtered according to the iTRAQ channels corresponding to OB samples (iTRAQ-117 and iTRAQ-118). For the differential protein abundance analysis, the normalized intensities of both iTRAQ-117 (wild-type) and iTRAQ-118 (*APP/PS1*) of each MS2 spectra were treated as paired samples, and the ratios between wild-type and *APP/PS1* were calculated based on these values. These ratios were log_2_ transformed and their mean, standard deviation (SD), and standard error of the mean (SEM) were calculated for each peptide. A one-sample t-test was used to assess whether the mean of log_2_(ratio) differed from zero for each peptide, adopting a 5% significance level.

### 2.3. Gene Orthology Assessment

The orthology relationship established and constantly updated by the National Center for Biotechnology Information (NCBI) was used to assign gene orthology from protein sequences of human and mouse customized protein datasets. The gene_orthologs.txt and gene2unigene.txt files were downloaded from the NCBI FTP site https://ftp.ncbi.nlm.nih.gov/gene/DATA/, accessed on 23 December 2020, and the HUMAN_9606_idmapping.dat and MOUSE_10090_idmapping.dat files were downloaded from the UniProt FTP site ftp://ftp.uniprot.org/pub/databases/uniprot/current_release/knowledgebase/idmapping/by_organism/, accessed on 23 December 2020. By using these files, the protein IDs of the customized protein sequence databases were parsed to gene IDs, and the orthologous genes between humans and mice were assigned based on the information present in the gene_orthologs.txt file (Figure 1). All computer programs and files used in this process can be found at https://github.com/Matheusdras/FindOrthologs, accessed on 26 March 2021.

### 2.4. Identification of AS Variants in Public RNA-Seq Data

The CC and OB RNA-Seq raw data were retrieved from the NCBI Gene Expression Omnibus with accession numbers GSE123496, GSE113524, GSE112352, and GSE106468 (https://www.ncbi.nlm.nih.gov/geo/, accessed on 17 February 2021). The raw reads were trimmed with Illumina sequencing default parameters using TrimGalore software [47]. Trimmed reads were aligned onto the human (UCSC hg38) and mouse (UCSC mm10) reference genomes using Hisat v.2 2.1.0 software [48], and the parameters used were adapted according to each library type and sequencing technology. Salmon v. 0.12.0 [49] was used to count the number of reads mapped to each transcript of interest. Transcripts per million (TPM) were calculated and used to compare the differences in transcript expression between OB and CC samples. Additionally, Cufflinks v.2.2.1 [50] was also used to measure the relative abundance of transcripts and was set with default parameters, according to library type and sequencing technology, and genome annotation files (GTF format) available in Ensembl Project version 102 [36].

### 2.5. Collection of Animal Samples

Ten-week-old adult male C57BL/6 SPF (specific pathogen-free) mice (*n* = 5) were obtained from the Animal Center of the Federal University of Minas Gerais, Belo Horizonte, Brazil. The mice were housed and maintained in a 12 h light/dark cycle and fed *ad libitum*. All procedures were performed following the guidelines for the humane use of laboratory animals and approved by the Institutional Animal Care and Use Committee established at the Federal University of Minas Gerais (protocol code 82/2021). Mice were deeply anesthetized with an intraperitoneal injection of a mixture containing ketamine (300 mg/kg BW) and xylazine (30 mg/kg BW), and the brains were rapidly removed. The OB (+3.56 mm to +4.28 mm) and CC (+0.74 mm to −0.10 mm) samples were isolated from either side of the brain and stored at −80 °C using a sterile stainless-steel blade and the stereotaxic coordinates relative to bregma.

### 2.6. Quantitative Reverse Transcription Polymerase Chain Reaction (RT-qPCR)

Total RNA was isolated using TRIzol reagent (Invitrogen), and 1 µg was reverse transcribed using the ImProm-II™ Reverse Transcription System (Promega) following the manufacturer’s instructions and anchored oligo (dT) 20 (IDT) primers. Spectrophotometry at a wavelength of 260 nm was used for RNA quantification in a NanoDrop^®^ 2000 (Thermo Fisher Scientific, Wilmington, Delaware, USA), and protein and other contaminants (phenol, polysaccharide, and salts) were estimated by ratios between 260/280 nm and 260/230 nm, respectively (Appendix A). The integrity of the total RNA was assessed by electrophoresis in 1% agarose gel in Tris-acetate-EDTA (TAE) buffer stained with SYBR^®^ Safe (Invitrogen) (Appendix A). cDNA (2 µL of 1:5 dilution) was also evaluated by electrophoresis in 1.2% agarose gel in 1X TAE buffer stained with SYBR^®^ Safe (Invitrogen) (Appendix A). It was then used as template in RT-qPCR with the SYBR Green Master Mix Detection System (Applied Technologies), with each primer at 5 or 10 pmol/µL (EXXTEND) (Appendix A) in a total volume of 10 µL. All reactions were run in technical triplicates in the 7900HT Fast Real-Time PCR System (Applied Biosystems), and a universal cycling program (1× 50 °C 2’; 95 °C 10’; 40× 95 °C 15’, 60 °C 1’) was used. Efficiency of 90–110% (Appendix A) was achieved in all reactions, and a single peak in the dissociation curve was observed (Appendix A). Whole-brain cDNA was used as a calibrator, and the reference gene Hprt1 (hypoxanthine phosphoribosyltransferase 1) was used to normalize the relative expression of target genes, which was calculated using the 2^−∆∆Ct^ method [51]. A two-sample t-test was used to assess whether the means of CC and OB 2^−∆∆Ct^ values of each AS variant were equal. The primer sequences are available in Appendix A, and a schematic representation of the spliced transcripts and their primer annealing positions is presented in Appendix A.

## 3. Results

### 3.1. Protein Sequence Database Setup

The human protein sequence database had 20,321 canonical proteoform sequences and 174,444 peptides of noncanonical proteoforms. The mouse protein sequence database had 17,033 canonical proteoform sequences and 163,056 peptides of noncanonical proteoforms.

### 3.2. Database Search and Orthology Assessment

Overall, after the database search of the CC datasets of healthy tissues, we identified 9086 peptides corresponding to 1909 non-redundant proteins for humans and 70,036 peptides corresponding to 9395 non-redundant proteins for mice. The database search in human OB datasets of healthy tissues had 53,379 peptides from 6563 non-redundant proteins, and 87,393 peptides from 10,384 non-redundant proteins in mice (Appendix A). We also found 1410 orthologous genes of canonical proteoforms and 3004 identical peptides (Figure 2A,B), as well as 27 orthologous genes of noncanonical proteoforms and 10 identical peptides across the different samples of both human and mouse brain tissues (Figure 2C,D) (Appendix A—Appendix A).

Among the noncanonical proteoforms of orthologous genes, we also identified proteotypic peptides, which are peptides that are exclusive to a protein sequence of the database [52] with identical sequences in human and mouse MS/MS datasets (Figure 3A). Besides sharing the same amino acid sequence, these peptides also possess a similar exon location on the mRNA exon/intron structure (Figure 3B). Appendix A shows all mass spectra of the identified peptides.

We also identified two peptides (ASSHSTDLMEAMAMGSVEASYK and CLAAALIVLTESGR) of the PKM1 proteoform, one peptide of the STXBP1a proteoform (WEVLIGSTHILTPTK) and one peptide of the LCRMP-1 proteoform (YGGMFAAVEGAYENK) in the isobaric labeled peptide (iTRAQ-8) data of the APP/PS1 model and its wild-type control. The peptide CLAAALIVLTESGR of PKM1 and WEVLIGSTHILTPTK of STXBP1a were more abundant in the APP/PS1 model than in the wild-type model (Figure 4). All the results from the iTRAQ data analysis are in Appendix A.

### 3.3. Identification of AS Variants at the Transcriptome Level

Publicly available RNA-Seq data were used to provide additional information about the expression of transcript variants of the *PKM/Pkm*, *STXBP1/Stxbp1*, *HNRNPK/Hnrnpk*, *CRMP1/Crmp1*, *CADM1/Cadm1*, and *PRKCB/Prkcb* genes in human and mouse CC and OB. We identified transcripts of all six genes of interest in both OB and CC tissues with TPM > 0 (Figure 5A) and FPKM > 0 (Appendix A). Among the AS variants, Pkm-202 (ENSMUST00000163694) was the most highly expressed in the two mouse brain regions. In human tissues, CADM1-212 (ENST00000542447) was the most highly expressed in the two brain regions. The AS variants of the *Cadm1*, *Crpd1*, *Hnrnpk*, *Pkm*, *Prkcb,* and *Stxbp* genes were amplified by RT-qPCR in both mouse CC and OB. AS variants of the Cadm1 and Prkcb showed higher expression in the OB (*p*-value ≤ 0.01) and AS variants of *Crmpd1* and *Hnrnpk* showed higher expression in the CC (*p*-value ≤ 0.05 and *p*-value ≤ 0.01, respectively) (Figure 5B).

## 4. Discussion

RNA-Seq and proteomics based on mass spectrometry are widely used technologies that provide molecular-level insights into neurodegenerative diseases such as AD [53,54,55,56]. However, few studies have used an orthology approach to understand the diversity of AS manifestations in the CC and OB in humans and mice. In this study, we built customized protein sequence databases for humans and mice, and used them to analyze publicly available data from CC and OB proteomes [44,45,46]. We were able to identify proteotypic peptides of orthologous AS proteoforms with identical sequences in both humans and mice (Figure 3A). Our findings are in line with previous studies, which identified the peptides of these proteoforms in different tissues [57,58,59]. All AS variants were confirmed at the transcript level by both RNA-Seq data (Appendix A) and RT-qPCR assays (Figure 4). Identifying these AS variants at the transcript and protein levels provides a fuller view of the transfer and expression of genetic information [60]. Furthermore, these AS variants were identified in both humans and mice, contributing to the reliability of our results and enabling testing in murine models. Since these AS proteoforms have high sequence similarity (Figure 3A) and conserved exon patterns (Figure 3B), they likely play similar roles in the CC and OB of both organisms. In this section, we present evidence from the literature regarding the functions of these genes in brain tissues and their role in brain disorders, as well as discussing the possible roles of these proteoforms in healthy and diseased brains.

*CRMP1* and its mouse ortholog encode the protein dihydropyrimidinase-related protein 1 (also known as collapsin response mediator protein 1). This protein is widely expressed in brain cells [61] and is involved in neuronal processes such as axonal growth [62,63] and dendritic spine maturation [64]. The differential expression of this gene in olfactory neuronal tissues has been studied in many human pathologies [65,66,67]. There are two AS proteoforms for this gene: one with a short first exon (isoform 1) and the other with a long first exon (LCRMP-1) [68]. Our analysis identified a proteotypic peptide of the proteoform LCRMP-1 in humans (UniProt ID: Q14194-2) and mice (Ensemble ID: ENSMUST00000114158) in MS/MS data of the OB. High expression levels of LCRMP-1 have been detected in non-small-cell lung carcinoma (NSCLC) [69] and are considered a promising potential biomarker for the diagnosis and prognosis of cancers [reviewed by [70]]. However, further studies are required to address its function in the neuronal context of healthy and diseased brains.

Cell adhesion molecule 1 is a protein encoded by the *CADM1* gene (also known as *SynCAM*) in humans; its ortholog in mice acts as a synaptic adhesion molecule in synaptic assembly [71,72]. The gene is alternatively spliced in the region between exons 7/11, producing two isoforms identified in humans and mice: SP3 (exons 7/11) and SP4 (exons 7/8/11) [73]. Several other isoforms have been reported in humans [74] and mice [75]. Our analysis identified a proteotypic peptide of human SP3 (UniProt ID: Q9BY67-5) and its mouse ortholog (UniProt ID: Q8R5M8-4) in the MS/MS data from a healthy OB. Currently, the function of SP3 in the brain context is unknown.

The *PRKCB* gene of humans (also known as *PKCβ*) encodes the protein kinase c beta, as does its mouse ortholog. Kinase c beta is a member of the serine/threonine protein kinase family and requires diacylglycerol (DAG) and Ca^2+^ for optimal performance [reviewed by [76]]. The gene can generate two proteoforms, PKCβI and PKCβII, produced via an alternative 3’ splicing site on the last exon of the gene (exon 17) [77]. Our analysis identified one peptide of PKCβII of humans (UniProt: P05771-2) and mice (UniProt: P68404-2) in the MS/MS data of healthy OB. *PRKCB* was thought to decrease β-amyloid aggregates, leading to neuronal protection [78,79]. Consistent with this view, a recent study has shown that a low expression level of *PRKCB* is a potential causative factor of AD and can accurately predict the onset of the disease. However, the role of each proteoform in the AD context has not been explored in detail [80]. Our findings may shed some light on the participation of PKCβII in AD.

The *HNRNPK* gene of humans and its mouse ortholog encode the heterogeneous nuclear ribonucleoprotein K, an RNA-binding protein. The gene plays a role in molecular neuronal processes like synaptic plasticity [81] and axonogenesis [82]. The disruption of *HNRNPK* proteins caused by de novo frameshift and splice donor site is associated with a syndrome characterized by intellectual disability, cardiac defects, and alterations in connective tissue and bones, demonstrating the importance of *HNRNPK* for the correct functioning of multiple organs and tissues [83]. This gene is alternatively spliced, producing four proteoforms (isoforms 1, 2, 3, and 4) through the inclusion or exclusion of exon 8 and alternative use of the 5’ splicing site of its last exon (exon 17) [84]. Our analysis identified one proteotypic peptide of isoform 3 in humans (UniProt ID: P61978-3) and mice (UniProt ID: P61979-3) from the MS/MS data of the OB. Although *HNRPNPK* proteoforms have been studied in cancer progression [85], their proteoforms in the healthy and diseased brain remain poorly understood. Therefore, our results should provide insight for future studies that aim to investigate its role in the central nervous system.

The *PKM* gene and its mouse ortholog encode the pyruvate kinase M protein, a glycolytic enzyme [86]. This gene can generate two AS proteoforms through the mutually exclusive selection of exons 9 (PKM1) and 10 (PKM2) [87]. PKM2 is generally more abundant in embryonic stem cells [88] and organs with low energy demands, while PKM1 is more abundant in skeletal muscle, heart, and brain tissues [89]. A recent study demonstrated that AD alters glycolytic gene expression, including *PKM*, on oligodendrocytes [53]. Martire and colleagues found PKM1 overexpression in AD transgenic TgCRND8 mice at three months of age [90]. Furthermore, recent studies have implicated *PKM* proteins as promising biomarkers for AD in cerebrospinal fluid [53,91,92,93]. Our analysis identified proteotypic peptides of PKM1 of humans (UniProt: P14618-2) and mice (UniProt: P52480-2) in MS/MS data from healthy CC and OB. We also found that the proteoform was more abundant in OB tissues of the *APP/PS1* mouse model than in the wild-type control. To the best of our knowledge, this is the first report of PKM1 overexpression in the OB of an AD mouse model. This indicates that the overexpression of PKM1 might be related to AD and that this peptide is a potential biomarker. Additionally, the proteoform PKM2 and isoform 3 (*HNRNPK* gene) were inferred in MS/MS data from a large set of oligodendrocytes in a previous study [39], corroborating our findings. Interestingly, the switch over from PKM2 to PKM1 is known to be mediated by *HNRNPK* proteins [94,95]. Thus, *PKM* and *HNRNPK* proteoform levels could also be investigated as AD indicators.

Syntaxin-binding protein 1, produced by the *STXBP1* gene (also known as *Munc-18*) in humans and its mouse ortholog, is a core protein for the docking of presynaptic vesicles on neuronal membranes [96]. This gene is alternatively spliced, with the inclusion or exclusion of exon 18, producing a long (STXBP1a) and a short (STXBP1b) variant, respectively. Previous reports described STXBP1a as more abundant in the OB than STXBP1b [97]. Our analysis identified a proteotypic peptide of STXBP1a of humans (UniProt ID: P61764-2) and mice (UniProt ID: O08599-2) in OB MS/MS data. The sequences of these proteoforms are 100% identical and probably have the same function in the OB of humans and mice (Figure 3A). A study conducted with postmortem brain tissues of patients with cognitive impairment and clinical dementia indicated that STXBP1a is associated with the likelihood of dementia in the elderly [98]. This gene was previously found to have a central role during the neurodegenerative process of OB [32]. We found that STXBP1a was more abundant in OB tissues of the *APP/PS1* model, indicating that it might be related to AD. In line with our findings, a study using a rat model showed that *STXBP1* proteoforms might form amyloid-like aggregates [99], an important feature associated with AD [100]. Additionally, *STXBP1* is widely related to different types of epileptic encephalopathies [101], schizophrenia, and bipolar syndrome [102,103].

The entorhinal cortex and hippocampus are brain structures that may exhibit neurodegeneration in early clinical manifestations of AD. As the disease progresses, neurodegeneration spreads throughout the frontal cortex and neocortex [104]. Currently, there are no publicly available data relating to these brain regions in healthy and diseased human and mouse brains, thus limiting our study.

On the other hand, publicly available data do exist for other brain regions related to AD. CC is a brain region frequently observed to be atrophied in AD patients, which is associated with cognitive impairment [105,106]. In addition, 90% of AD patients demonstrate olfactory dysfunction and OB deregulation in the early stages of the disease, a symptom which is a reliable premature marker of AD [107]. Analyzing data from brain structures with classical AD-related neurodegeneration could unveil other orthologous AS proteoforms that could be tested in murine models.

Taken together, our strategy of using orthology data to improve the search for AS proteoforms produced encouraging results. Our approach may open a promising avenue for studies of orthologous AS proteoforms in humans and mice, which can assist human neurodegenerative studies using mouse models for other diseases. Our findings provide insights on the expression of six orthologous AS proteoforms at the transcript and protein levels between human and mouse brains, with support from the literature.

## 5. Conclusions

Our computational strategy to search for orthologous, alternatively spliced variants in humans and mice revealed a set of proteoforms differentially expressed at the mRNA level and with differential abundancy at the proteome level using human and mouse brain data as a proof of concept. We believe that our results contribute to a more comprehensive understanding of CC and OB splicing diversity. We hope that our findings encourage other research groups to test the clinical applications of these AS variants, particularly PKM1 and STXBP1a, using targeted proteomics technologies such as selected reaction monitoring (SRM) and multiple reaction monitoring (MRM). To our knowledge, this is the first report of PKM1 and STXBP1a overexpression in the OB of an AD mouse model, which should be further examined in order to characterize the isoform’s role in healthy and diseased brain tissues.

## Figures and Tables

**Figure 1 cells-10-01583-f001:**
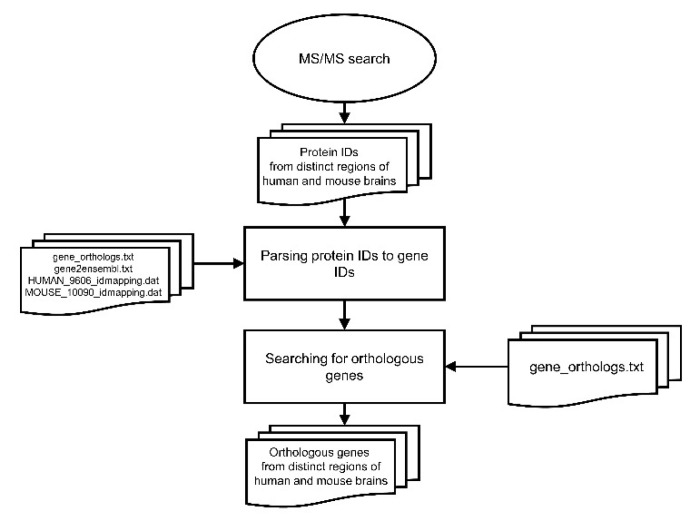
Flowchart for ortholog assessment.

**Figure 2 cells-10-01583-f002:**
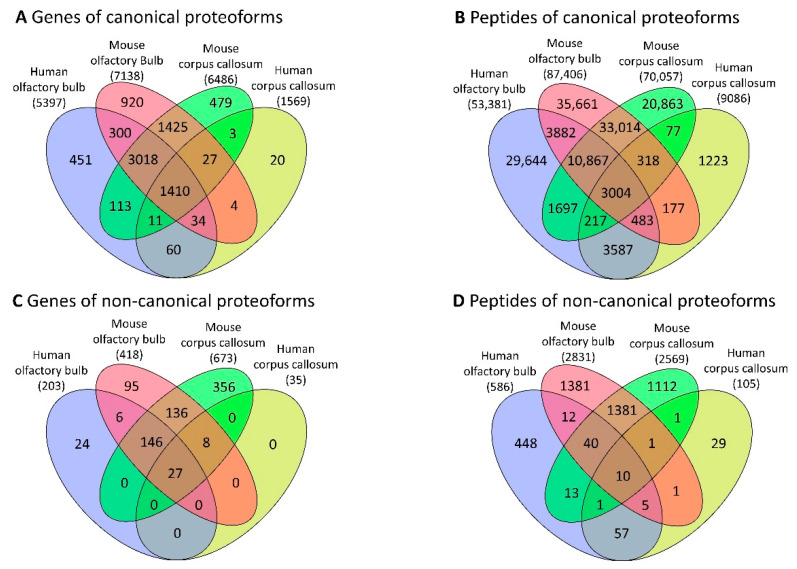
Distribution of genes and peptides across the corpus callosum and olfactory bulb in humans and mice. (**A**) Genes of canonical proteoforms with orthologous relationships between humans and mice or those shared across brain tissues are represented at intersections; (**B**) genes of noncanonical proteoforms with orthologous relationships between humans and mice or those shared across brain tissues are represented at intersections; (**C**) peptides of canonical proteoforms with identical sequences across human and mouse and brain regions are represented at intersections; (**D**) peptides of noncanonical proteoforms with identical sequences across human and mouse and brain tissues are represented at intersections.

**Figure 3 cells-10-01583-f003:**
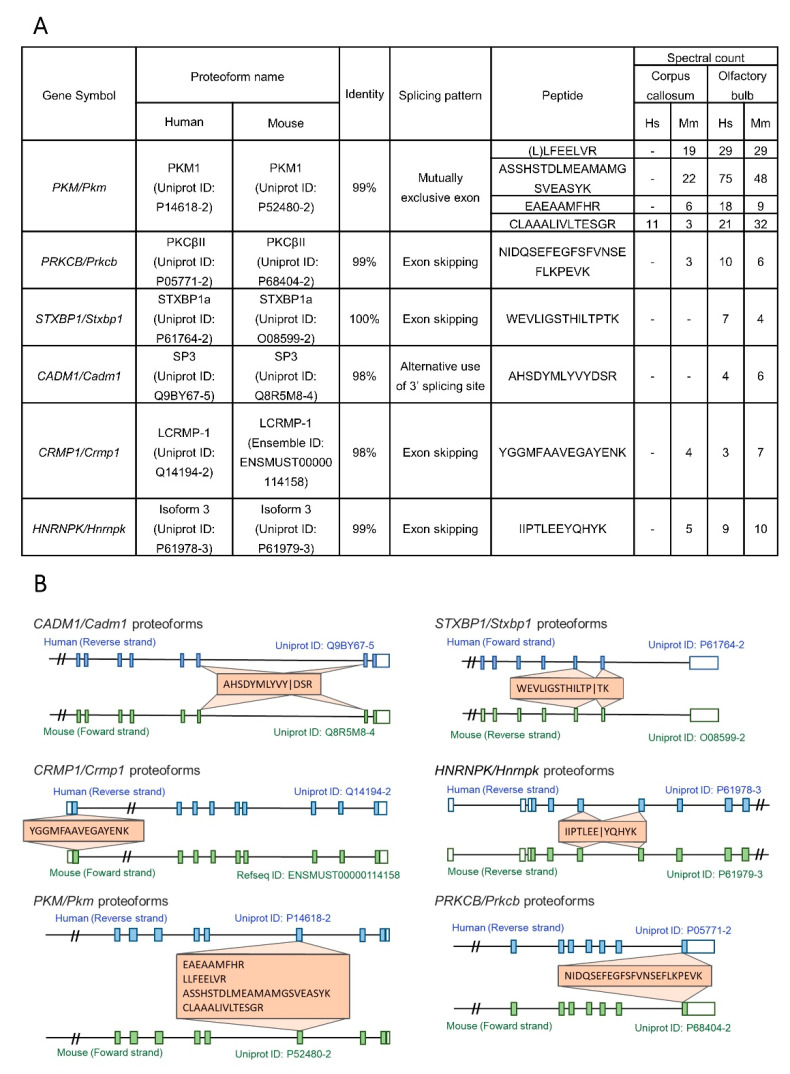
Exons and shared peptides of the orthologous alternatively spliced proteoforms. (**A**) Table with general information about the human (Hs) and mouse (Mm) proteoforms: gene symbol, proteoform name, identity, splicing pattern, identified peptides, and their spectral count in each brain region. (**B**) Schematic representation of the spliced transcripts of the genes CADM1/Cadm1; STXBP1/Stxbp1; CRMP1/Crmp1; HNRNPK/Hnrnpk; PKM/Pkm; PRKCB/Prkcb and the exon location of their identified peptides. Human proteoforms are represented in blue and mouse proteoforms are represented in green. Exons are represented by full (coding sequences) and empty (UTRs) boxes, while introns are represented by lines. Each peptide is represented with orange boxes. The two slashes in the intronic region indicate a fragment of mRNA structure not represented in the schematic.

**Figure 4 cells-10-01583-f004:**
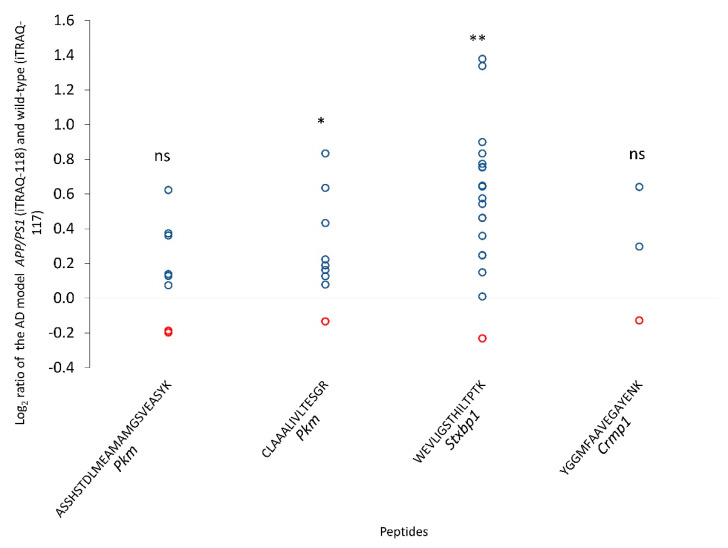
Log_2_ ratio of the normalized intensities of the AS proteoform peptides (genes *Pkm*, *Stxbp1* and *Crmp1*). Samples more abundant in the wild-type (log_2_ ratio < 0) are represented as red dots and samples more abundant in the AD model APP/PS1 (log_2_ ratio > 0) are shown as blue dots. One-sample t-test; not significant (ns), *p*-value < 0.05 (*) and *p*-value < 0.01 (**).

**Figure 5 cells-10-01583-f005:**
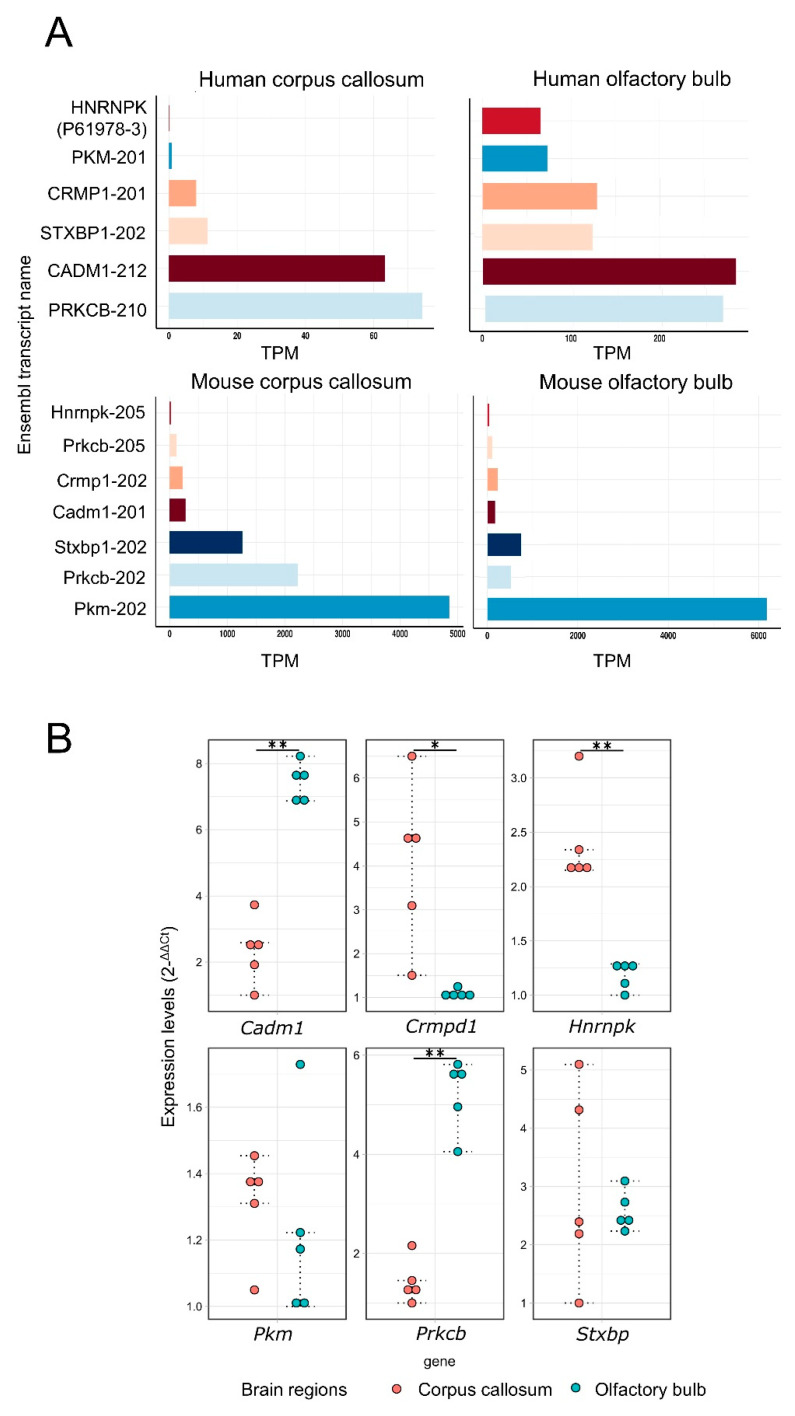
RNA-Seq expression levels for the orthologous alternatively spliced proteoforms selected for mRNA experimental validation. (**A**) Comparative analysis of transcript expression levels in transcripts per million (TPM) across the corpus callosum and olfactory bulb in mice and humans. The Y-axis shows the expression of transcripts in TPM, and the X-axis shows the different transcripts. (**B**) Transcript expression levels (2^−∆∆Ct^) of the AS variants of the genes Cadm1, Crmpd1, Hnrnpk, Pkm, Prkcb, and Stxbp in the mouse CC and OB. Two-sample t-test; *p*-value ≤ 0.05 (*) and *p*-value ≤ 0.01 (**).

**Table 1 cells-10-01583-t001:** Mass spectrometry data description.

Data Information	Human		Mouse	
Brain region	Corpus callosum	Olfactory bulb	Corpus callosum	Olfactory bulb
Database ID	PXD000547	PXD005629	PXD001250	
Reference	[44]	[45]	[46]	
Mass spectrometer	LTQ-Orbitrap XL	Orbitrap fusion tribrid	Q exactive	
Number of samples	2	3	3	4

## Data Availability

Publicly available datasets were analyzed in this study. These data can be found at: http://proteomecentral.proteomexchange.org/cgi/GetDataset?ID=PXD000547, Accessed on 10 December 2020; http://central.proteomexchange.org/cgi/GetDataset?ID=PXD005629, Accessed on 10 December 2020; http://proteomecentral.proteomexchange.org/cgi/GetDataset?ID=PXD001250, Accessed on 10 December 2020; http://proteomecentral.proteomexchange.org/cgi/GetDataset?ID=PXD003312, Accessed on 10 December 2020; https://www.ncbi.nlm.nih.gov/geo/GSE123496, Accessed on 17 February 2021; https://www.ncbi.nlm.nih.gov/geo/GSE113524, Accessed on 17 February 2021; https://www.ncbi.nlm.nih.gov/geo/GSE112352, Accessed on 17 February 2021; https://www.ncbi.nlm.nih.gov/geo/GSE106468, Accessed on 17 February 2021. All pipelines, command lines and programs used are available at https://github.com/Matheusdras/Make_Custom_Database and https://github.com/Matheusdras/FindOrthologs.

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
