# Peer review of "Proteogenomics Reveals Orthologous Alternatively Spliced Proteoforms in the Same Human and Mouse Brain Regions with Differential Abundance in an Alzheimer’s Disease Mouse Model"

_cells, 2021, doi:10.3390/cells10071583_

Round 1

Reviewer 1 Report

The authors analyzed protein database to compare proteotypic peptides between human and mouse and claimed found that Alternative splicing (AS) isoforms of several genes, further performed RT-qPCR to confirmed. Among them, PKM1 and STXBP1a were found to be highly abundant in AD mouse. The manuscript does not include sufficient contents and not fit to publication in Cells.

Major points.

  1. Protein Orthology differences in human and mouse are well defined in numerous study. It is surprising that they found only a few orthology. Peptide differences should be shown with western blotting if antibodies available. Otherwise, the findings are meaningless.
  2. AS isoforms should also be shown with RT-PCR in addition to RT-pPCR to clearly show the differences in sizes of different isoforms.
  3. They found the AS isoforms of a few genes. WHat is the biological siginificance of these isoforms?

Reviewer 2 Report

1. This manuscript is well organized but still needs a proofreading in English grammar and some spelling.
2. The format of the manuscript can be improved for better understanding. The Abstract and Discussion sections should be simplified but more previous published AD-related pathological OMICS references must be supplied in the manuscript.
3. The statistics p value should be supplied in the Figure 5b qRT-PCR data.
4. The major defect of the research design is that both corpus callosum (CC) and olfactory bulb (OB) are not the major pathological damages of AD. The author should discuss this defect in Discussion section.

Reviewer 3 Report

This papers by Gomes da Silva and collaborators deals with an analysis of the differential expression of proteoforms highly involved in AD in two parts of the brain (corpus callosum and olfactory bulb) and in two species (human and mouse). All the relevant analytical method(s) are bioinformatics-based and involve analysis of datasets available in public repositories.

The paper very interesting and well-written. The results refer to the differential expression of proteoforms in a limited number of genes but all directly or indirectly related to AD (as from the literature). The paper can be useful for all the people who want to approach differential expression of genes in tissues.

I have no objections on this paper. I only found that before being published a thorough revision of the References section is required since many references are not properly formatted.

Round 2

Reviewer 1 Report

The manuscript has been substantially modified and now is acceptable to Cells.

Reviewer 2 Report

The manuscript had been improved and can be accepted for publication.